# Comparative analysis of two *Caenorhabditis elegans* kinesins KLP-6 and UNC-104 reveals a common and distinct activation mechanism in kinesin-3

**Tomoki Kita[1], Kyoko Chiba[2], Jiye Wang[3], Atsushi Nakagawa[3], Shinsuke Niwa[1,2]***

[1]Graduate School of Life Sciences, Tohoku University, Sendai, Japan; [2]Frontier Research Institute for Interdisciplinary Sciences (FRIS), Tohoku University, Sendai, Japan; [3]Institute for Protein Research, Osaka University, Osaka, Japan

**\*For correspondence:**
shinsuke.niwa.c8@tohoku.ac.jp

**Competing interest:** The authors declare that no competing interests exist.

**Abstract** Kinesin-3 is a family of microtubule-dependent motor proteins that transport various cargos within the cell. However, the mechanism underlying kinesin-3 activations remains largely elusive. In this study, we compared the biochemical properties of two *Caenorhabditis elegans* kinesin-3 family proteins, KLP-6 and UNC-104. Both KLP-6 and UNC-104 are predominantly monomeric in solution. As previously shown for UNC-104, non-processive KLP-6 monomer is converted to a processive motor when artificially dimerized. We present evidence that releasing the autoinhibition is sufficient to trigger dimerization of monomeric UNC-104 at nanomolar concentrations, which results in processive movement of UNC-104 on microtubules, although it has long been thought that enrichment in the phospholipid microdomain on cargo vesicles is required for the dimerization and processive movement of UNC-104. In contrast, KLP-6 remains to be a non-processive monomer even when its autoinhibition is unlocked, suggesting a requirement of other factors for full activation. By examining the differences between KLP-6 and UNC-104, we identified a coiled-coil domain called coiled-coil 2 (CC2) that is required for the efficient dimerization and processive movement of UNC-104. Our results suggest a common activation mechanism for kinesin-3 family members, while also highlighting their diversification.

## eLife assessment

This study explores the activation mechanisms of members of the kinesin-3 family, demonstrating common and unique regulation modes with **solid** evidence. The findings make for **valuable** contributions to the field of kinesin activation and regulation.

## Introduction

Cellular morphogenesis depends on intracellular transport. Kinesins, also known as kinesin superfamily proteins (KIFs), are microtubule-dependent molecular motors that are essential for intracellular transport and cell division (*Hirokawa et al., 2009*). Among kinesins, the kinesin-3 family is mainly involved in intracellular transport, including axonal transport of synaptic materials, lysosomal transport, mitochondrial transport, and intraflagellar transport of mechanoreceptor complexes (*Chiba et al., 2023*; *Guardia et al., 2016*; *Morsci and Barr, 2011*; *Nangaku et al., 1994*; *Okada et al., 1995*).

The functions and regulation of the kinesin-3 family have been elucidated through genetic studies in *Caenorhabditis elegans* (*C. elegans*) (*Chiba et al., 2019*; *Cong et al., 2021*; *Hall and Hedgecock, 1991*; *Kumar et al., 2010*; *Niwa et al., 2016*; *Niwa et al., 2017*; *Otsuka et al., 1991*; *Wu*

**Figure 1.** Full-length KLP-6 shows diffusive motion on microtubules. (**A**) Schematic drawing of the domain organization of KLP-6 motor protein and the full length of KLP-6 fused with sfGFP (KLP-6FL). NC, neck coiled-coil domain. CC1, coiled-coil 1 domain. FHA, forkhead-associated domain. CC2, coiled-coil 2 domain. MBS, membrane-associated guanylate kinase homolog (MAGUK)-binding stalk domain. MATH, meprin and TRAF homology domain. Calculated molecular weight (MW) is shown at the right side. (**B**) Size exclusion chromatography of KLP-6FL. The SDS-PAGE of the elution fractions is shown beneath the profile. Asterisks indicate fractions used for mass photometry and single molecule assays. The void volume of the column is indicated. Asterisks indicate fractions that are used for mass photometry and total internal reflection fluorescent (TIRF) assays. Number shown at the left side indicates a molecular weight standard. (**C**) Mass photometry of KLP-6FL. Histogram shows the particle count of KLP-6FL at 20 nM. The line shows a Gaussian fit (mean ± standard deviation [SD]: 124±15.5 kDa). (**D**) Representative kymographs showing the motility of 5 pM KLP-6FL in the presence of 2 mM ATP. Note that KLP-6FL shows only one-dimensional diffusion on microtubules but does not show any processive runs. Horizontal and vertical bars shows 10 µm and 10 s, respectively.

The online version of this article includes the following source data and figure supplement(s) for figure 1:

**Source data 1.** Original file for the SDS-PAGE analysis in *Figure 1B* (KLP-6FL).

**Source data 2.** PDF containing *Figure 1B* and a original scan of the relevant SDS-PAGE analysis (KLP-6FL) with a highlighted band.

**Figure supplement 1.** Microtubule gliding assays using KLP-6.

**Figure supplement 1—source data 1.** Velocities in the microtubule-gliding assays.

---

*et al., 2013*; *Zheng et al., 2014*). *C. elegans* has three members of the kinesin-3 family: UNC-104, KLP-4, and KLP-6. UNC-104 transports synaptic vesicle precursors, mature synaptic vesicles, and pre-synaptic membrane proteins in the axon (*Hall and Hedgecock, 1991*; *Oliver et al., 2022*; *Otsuka et al., 1991*). Its mammalian orthologs, KIF1A and KIF1Bβ, also play a role in the axonal transport of synaptic materials (*Niwa et al., 2008*; *Okada et al., 1995*; *Zhao et al., 2001*). Because mutations in KIF1A and KIF1Bβ have been associated with congenital disorders (*Boyle et al., 2021*; *Budaitis et al., 2021*; *Klebe et al., 2012*; *Zhao et al., 2001*), *C. elegans* has been used to study the molecular mechanisms of pathogenesis (*Anazawa et al., 2022*; *Chiba et al., 2023*; *Lam et al., 2021*). KLP-4, on the other hand, is responsible for the transport of glutamate receptor called GLR-1 in the dendrite. The transport of GLR-1 is significantly reduced in *klp-4* mutant worms (*Monteiro et al., 2012*). KLP-6 is an invertebrate specific kinesin-3 and transports a mechanosensory receptor complex in male cilia (*Morsci and Barr, 2011*; *Peden and Barr, 2005*). This mechanoreceptor complex consists of LOV-1 and PKD-2, orthologs of mammalian polycystin-1 and polycystin-2 (*Barr et al., 2001*). In human, muta-tions in *PKD-1* or *PKD-2* gene which encodes polycystin-1 or polycystin-2 cause autosomal dominant

polycystic kidney disease (*Hughes et al., 1995*; *Mochizuki et al., 1996*). In worms, mutations in *klp-6* gene lead to the reduction of LOV-1 and PKD-2 from the cilia and the male infertility because the mechanoreceptor complex is essential for male worms to locate the hermaphrodite vulva during mating behavior (*Peden and Barr, 2005*).

Among these kinesin-3 family members, extensive biochemical and structural analyses have been conducted on UNC-104 and KLP-6 (*Al-Bassam et al., 2003*; *Klopfenstein et al., 2002*; *Tomishige et al., 2002*; *Wang et al., 2022*). It has been proposed that these motors are regulated by a monomer-to-dimer conversion mechanism (*Al-Bassam et al., 2003*; *Tomishige et al., 2002*; *Wang et al., 2022*). UNC-104 dimers, which are stabilized by fusing with a coiled-coil domain of kinesin-1, exhibit plus-end directed movement on microtubules, whereas purified UNC-104 monomers show only one-dimensional diffusion (*Soppina et al., 2014*; *Tomishige et al., 2002*). UNC-104 requires a high concentration (at least the range of 1–7 µM) for dimerization (*Tomishige et al., 2002*). Furthermore, UNC-104 mini-motors, generated by deleting most of the stalk domains, have been shown to be enriched in phosphatidylinositol 4, 5-bisphosphate (PIP2) microdomains on artificial liposomes (*Klopfenstein et al., 2002*). As a result, the mini-motor can efficiently transport PIP2-carrying vesicles in vitro. These results collectively suggested that the non-processive UNC-104 monomer is accumulated to high concentrations and converted into an active dimer at PIP2 microdomains on the cargo membrane (*Klopfenstein et al., 2002*; *Tomishige et al., 2002*). Cryo-electron microscopy analysis of UNC-104 and the X-ray structure of full-length KLP-6 have revealed the autoinhibited state of kinesin-3 (*Al-Bassam et al., 2003*; *Ren et al., 2018*; *Wang et al., 2022*). These structures suggest that stalk domains, including coiled-coil 1 (CC1), forkhead-associated (FHA), and coiled-coil 2 (CC2) domains, bind to the motor domain (MD) and the neck coiled-coil (NC) domain and inhibit the microtubule-dependent motor activity of kinesin-3 (*Ren et al., 2018*; *Wang et al., 2022*).

Based on the genetic screening and the crystal structure, point mutations that disrupt the autoinhibition of UNC-104 and KLP-6 have been identified. These prior studies have indirectly detected the activation of UNC-104 and KLP-6 by observing the localization of cargo vesicles, measuring ATPase activity or observing tip accumulation in neuronal cells (*Cong et al., 2021*; *Niwa et al., 2016*; *Wagner et al., 2009*; *Wang et al., 2022*; *Wu et al., 2013*). However, we still lack insights into oligomeric states of UNC-104 and KLP-6 at submicromolar concentrations. Without direct visualization of purified UNC-104 and KLP-6 itself at low nanomolar concentrations, it is difficult to determine whether UNC-104 and KLP-6 are capable of dimerization on their own upon autoinhibition release or whether other cellular factors are required for their dimerization. Furthermore, confirming that UNC-104 and KLP-6 motors form dimers at physiological concentrations is necessary, since the concentration of most cellular proteins is at the range of nanomolar (*Wühr et al., 2014*).

In this study, we conducted mass photometry assays and single molecule motility assays to investigate the oligomeric state and motility of two purified *C. elegans* kinesin-3 motors, KLP-6 and UNC-104. Our results demonstrate that UNC-104, but not KLP-6, can form a dimer on its own in solution at nanomolar concentrations when autoinhibition is released. We also found that a previously unexplored coiled-coil domain, called the CC2 domain, is essential for the formation of UNC-104 dimers at nanomolar concentrations. The CC2 domain was essential for the processive movement of UNC-104. Unlocking of the autoinhibition alone was sufficient to induce the dimerization and activation of UNC-104 without the binding with cargo vesicles.

## Results

### Full-length KLP-6 is a monomeric motor

Although the full-length structure of KLP-6 has been solved previously (*Wang et al., 2022*), its biochemical properties have not been thoroughly examined. Here, we have characterized the motile properties of a recombinant KLP-6 construct consisting of full-length KLP-6 (aa 1–928) with a C-terminal green fluorescent protein (KLP-6FL) (*Figure 1A*). In a size exclusion chromatography (SEC), KLP-6FL was eluted from a single peak (*Figure 1B*). Mass photometry confirmed that KLP-6FL was monomeric in solution (*Figure 1C*), as in the previous study (*Wang et al., 2022*). We found purified KLP-6FL was an active microtubule-dependent motor in a microtubule gliding assay (*Figure 1—figure supplement 1*). First, KLP-6-FL was attached on the glass surface using an anti-GFP antibody. In this condition, the velocity of the movement in the gliding assay (52.4±33.9 nm/s, mean ± standard

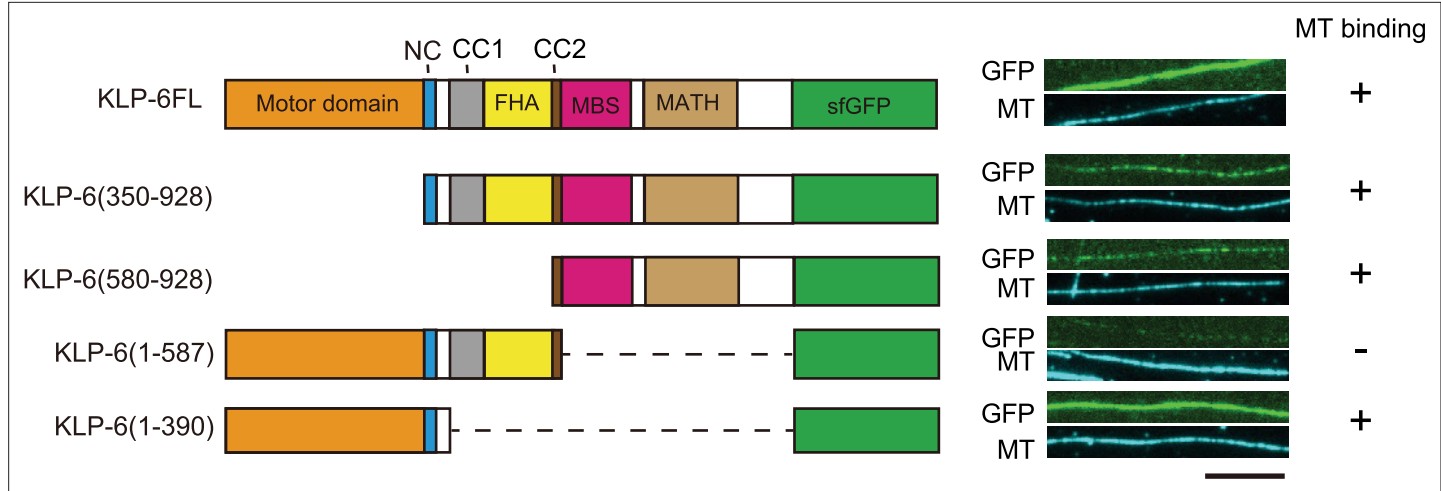

**Figure 2.** KLP-6 has a second microtubule binding domain. Schematic drawing of the domain organization of KLP-6 analyzed and representative total internal reflection fluorescent (TIRF) images showing the bindings of GFP fused KLP-6 deletion mutant proteins (GFP, green) to microtubules (MT, cyan). The tail domain alone binds to microtubules. Bar, 10 μm.

The online version of this article includes the following source data and figure supplement(s) for figure 2:

**Figure supplement 1.** Purified KLP-6 and its deletion mutants do not show processive runs.

**Figure supplement 1—source data 1.** Landing rate and dwell time of KLP-6 mutants.

deviation [SD], n=115 microtubules) was approximately 14-fold slower than that of KLP-6::GFP in cilia (0.72±0.18 μm/s) (*Morsci and Barr, 2011*; *Figure 1—figure supplement 1A, B*). Second, KLP-6FL was attached on the glass surface directly. This led to an increased velocity in the gliding assay (154.9±20.3 μm/s, mean ± SD, n=127 microtubules) (*Figure 1—figure supplement 1A, B*), implying that such attachment activates the motor in the gliding assay. We next examined KLP-6FL processivity by imaging single GFP-labeled molecules using a total internal reflection fluorescent (TIRF) microscope. Our results showed that KLP-6FL bound to microtubules and exhibited one-dimensional diffusion, but rarely showed directional movement (*Figure 1D*).

### KLP-6 has a second microtubule binding domain in the tail

While we observed KLP-6FL bound to microtubules and showed diffusion on microtubules (*Figure 1D*), the structure of KLP-6 shows that the stalk domains of KLP-6 cover its MD and prevent the MD from binding to microtubules (*Wang et al., 2022*). This raised a possibility that the tail domain of KLP-6, rather than the MD, bind to microtubules in the autoinhibited state. To test this hypothesis, we purified deletion mutants of KLP-6 and examined their association with microtubules. We found that KLP-6 (350–928), which lacks the MD, and KLP-6 (580–928), which lacks the MD-NC-CC1-FHA domains, bound to microtubules (*Figure 2*). Inversely, KLP-6 (1–587), lacking the MBS and MATH domains, rarely bound to microtubules. However, KLP-6 (1–390), which lacks the CC1-FHA-CC2 domains, was able to bind to microtubules. Note that none of the KLP-6 proteins showed processive movement on microtubules (*Figure 2—figure supplement 1A–C*). These suggest that KLP-6 has a second microtubule binding domain in the tail domains.

### Dimerization prompts processive movement of the KLP-6 motor

KLP-6 (1–390) but not KLP-6 (1–587), bound to microtubules (*Figure 2*). This result is consistent with previous findings showing that the CC1-FHA-CC2 domains can inhibit the motor activity in kinesin-3 (*Hammond et al., 2009*; *Niwa et al., 2016*; *Wang et al., 2022*). Although KLP-6 (1–390) did not show any processive runs on microtubules in the TIRF assay (*Figure 2—figure supplement 1A*), we found KLP-6 (1–390) is an active microtubule-dependent motor in a microtubule gliding assay (*Figure 1—figure supplement 1C*). The motor was attached on the glass surface using the anti-GFP antibody and the velocity of microtubule gliding was 120.3±15.3 nm/s (mean ± SD, n=180 microtubules). For kinesin-3 motors, it has been proposed that dimerization is required to achieve processive runs on

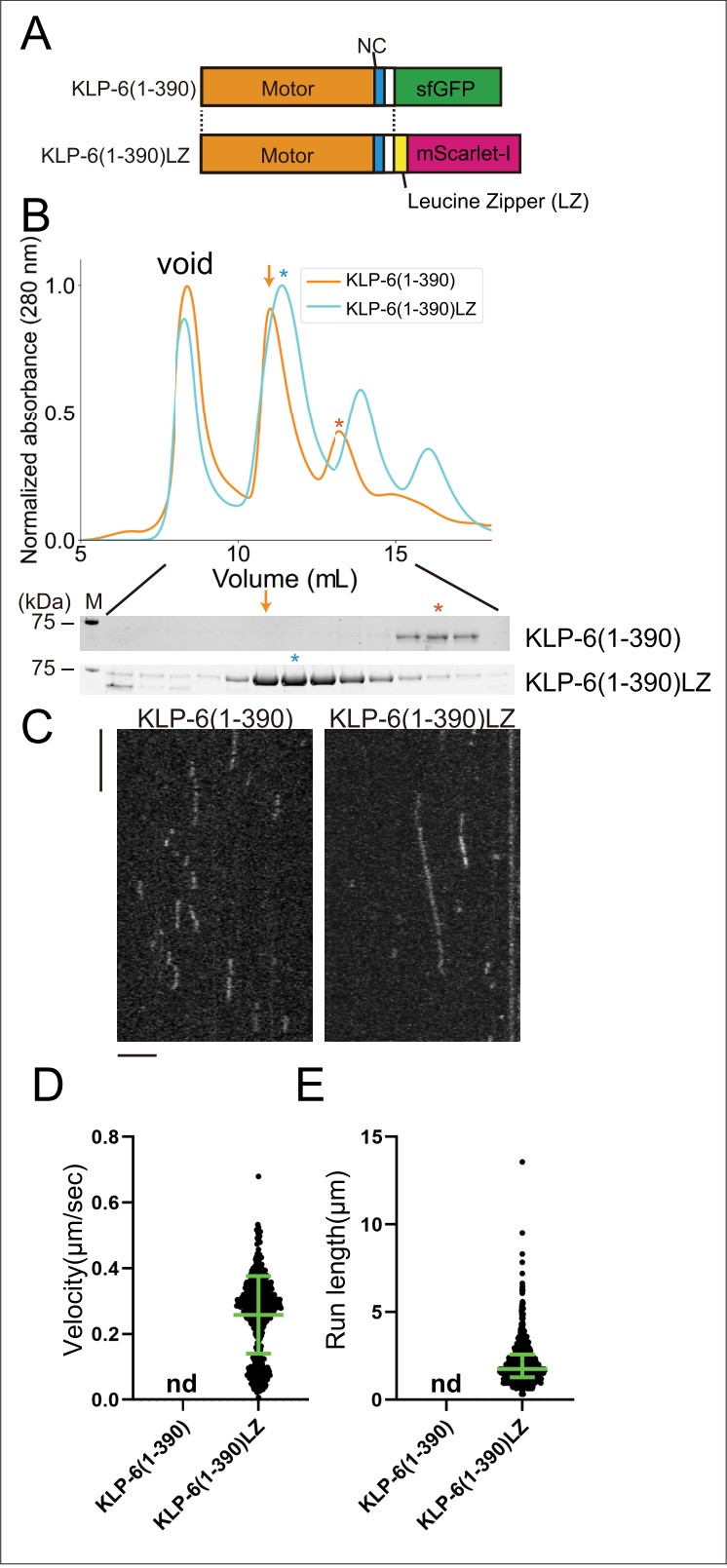

**Figure 3.** KLP-6 is converted to a processive motor upon dimerization. (**A**) Schematic drawing of the domain organization of KLP-6 (1–390) and KLP-6 (1–390)LZ. (**B**) Size exclusion chromatography of KLP-6 (1–390) (orange) and KLP-6 (1–390)LZ (cyan). The SDS-PAGE of the elution fractions are shown beneath the profiles. The number shown at the left side indicates molecular weight standard. Orange and cyan asterisks indicate fractions used

*Figure 3 continued on next page*

*Figure 3 continued*

for single molecule assays. Note that the fraction indicated by an orange arrow does not contain KLP-6 (1–390) protein. (**C**) Representative kymographs showing the motility of 5 pM KLP-6 (1–390) and KLP-6 (1–390)LZ in the presence of 2 mM ATP. Horizontal and vertical bars show 5 s and 5 µm, respectively. (**D**) Dot plots showing the velocity of KLP-6 (1–390) and KLP-6 (1–390)LZ. Each dot shows a single datum point. Green bars represent mean ± SD, n=642 for KLP-6 (1–390)LZ. n.d., no directional movement was detected in KLP-6 (1–390). (**E**) Dot plots showing the run length of KLP-6 (1–390) and KLP-6 (1–390)LZ. Each dot shows a single datum point. Green bars represent median value and interquartile range. n=642 for KLP-6 (1–390)LZ. n.d., no directional movement was detected in KLP-6 (1–390).

The online version of this article includes the following source data for figure 3:

**Source data 1.** Velocity and run length of KLP-6 (1–390)LZ.

**Source data 2.** Original file for the SDS-PAGE analysis in *Figure 3B* (KLP-6(1-390) and KLP-6(1-390)LZ).

**Source data 3.** PDF containing *Figure 3B* and original scans of the relevant SDS-PAGE analysis (KLP-6(1-390) and KLP-6(1-390)LZ) with highlighted bands.

microtubules (*Soppina et al., 2014*). Therefore, we dimerized KLP-6 (1–390) using the LZ domain of GCN4 (*Soppina et al., 2014*; *Figure 3A and B*). As a result, we found KLP-6 (1–390)LZ moved on microtubules processively in the TIRF assay (*Figure 3C–E*). The velocity was 0.26±0.12 µm/s (mean ± SD, n=642 molecules). These results suggest that KLP-6 is an inactive monomer and requires dimerization to perform directional movement on microtubules.

## KLP-6 remains to be a non-processive monomer when the autoinhibition is relieved

In a previous study, UNC-104(1–653) was shown to be a monomeric and inactive protein, also known as U653 (*Tomishige et al., 2002*). KLP-6 (1–587) has a similar domain architecture to UNC-104(1-653) (*Figure 4A* and *Figure 4—figure supplement 1*) and was an inactive motor. To investigate the activation mechanisms of kinesin-3, we further analyzed the KLP-6 (1–587) and UNC-104(1–653) motors, respectively. *Wang et al., 2022* identified mutations that disrupt the autoinhibition of KLP-6 based on its autoinhibited structure (*Wang et al., 2022*). It was demonstrated that these mutations activate KLP-6 in cultured cells (*Wang et al., 2022*). Among mutations analyzed in their study, KLP-6(D458A), a mutation that disrupts the intramolecular interaction between the FHA domain and MD, has the strongest effect on the activation of KLP-6 (*Figure 4A* and *Figure 4—figure supplement 2*; *Wang et al., 2022*). Therefore, we introduced the D458A mutation into KLP-6 (1–587) to create a mutant protein, KLP-6 (1–587)(D458A). KLP-6 (1–587)(D458A) exhibited similar properties to wild-type KLP-6 (1–587) in the SEC analysis (*Figure 4B*). Subsequent mass photometry confirmed that both KLP-6 (1–587) and KLP-6 (1–587)(D458A) were predominantly monomeric in solution (*Figure 4C*). Under the condition of the microtubule gliding assay with the anti-GFP antibody, KLP-6 (1–587) (116.7±17.9 nm/s, mean ± SD, n=107 microtubules) as well as KLP-6 (1–587)(D458A) (114.3±18.9 nm/s, mean ± SD, n=103 microtubules) exhibited comparable activity to KLP-6 (1–390) (120.3±15.3 nm/s, mean ± SD, n=180 microtubules) even without attaching the motor to the surface directly (*Figure 1—figure supplement 1D*). Despite these similar properties between KLP-6 (1–587) and KLP-6 (1–587)(D458A) in the gliding assay, KLP-6 (1–587)(D458A), but not wild-type KLP-6 (1–587), frequently bound to microtubules in the TIRF assay (*Figure 4D–F*). However, KLP-6 (1–587)(D458A) only exhibited one-dimensional diffusion and did not show any processive runs on microtubules (*Figure 4D*). Considering that dimeric KLP-6 (1–390)LZ, but not monomeric KLP-6 (1–390), are processive, these observations indicate that the autoinhibition of KLP-6 (1–587)(D458A) is relieved but that KLP-6 (1–587)(D458A) still cannot form processive dimers.

## Relieving autoinhibition in UNC-104 is sufficient to induce processive movement

Our previous work has identified mutations in the CC1 domain of UNC-104, such as UNC-104(E412K), that activate axonal transport of synaptic vesicle precursors (*Niwa et al., 2016*). Autoinhibited KLP-6 and KIF13B structures suggest that the E412K mutation in the UNC-104 disrupts the autoinhibition of UNC-104 (*Figure 4—figure supplement 2*) similarly to the KLP-6(D458A) mutation (*Ren*

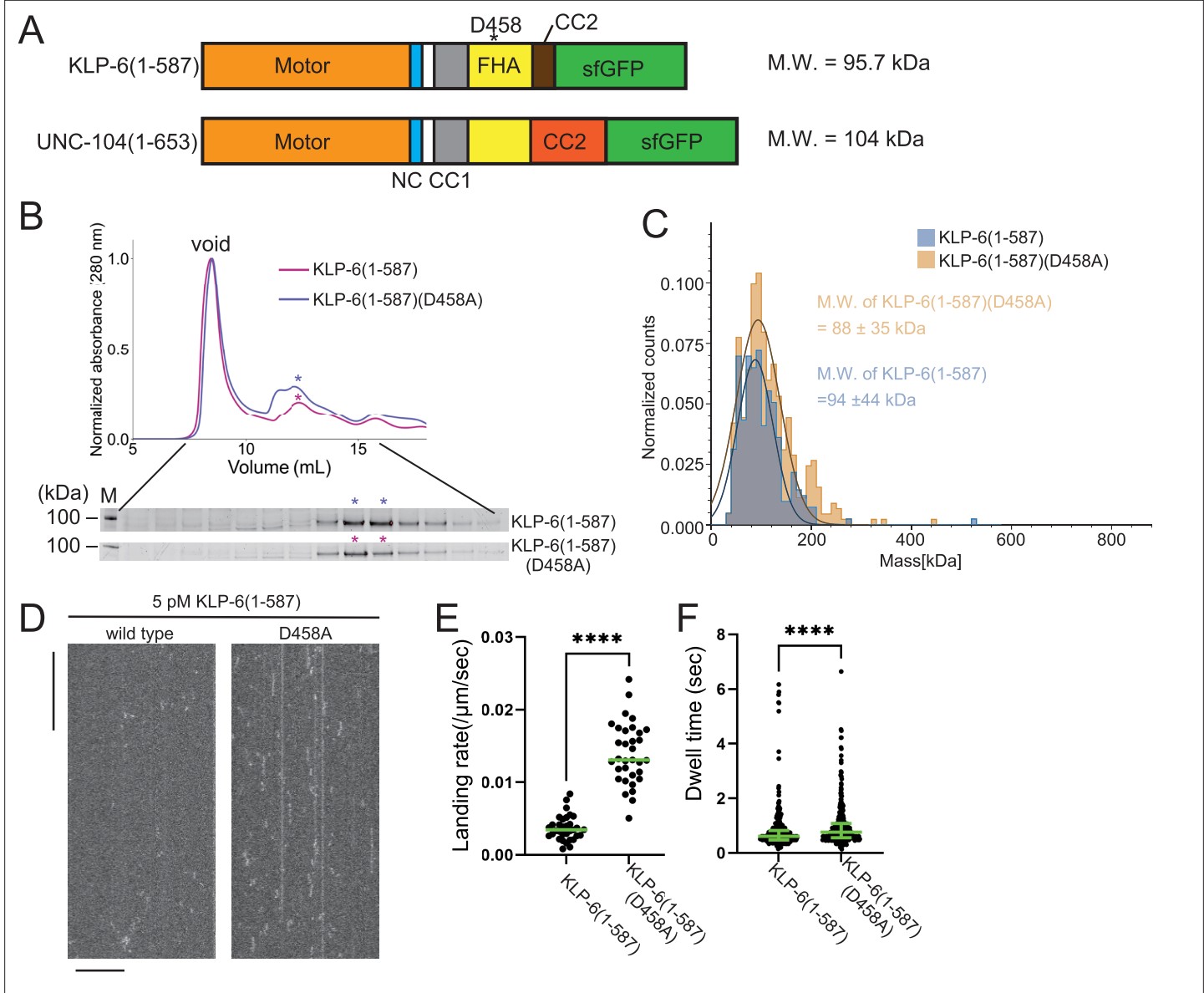

**Figure 4.** KLP-6 is a constitutive monomer. (**A**) Schematic drawing of the domain organization of KLP-6 (1–587) and UNC-104(1–653). Calculated molecular weight is written at the right side. (**B**) Size exclusion chromatography of KLP-6 (1–587) (plum) and KLP-6 (1–587)(D458A) (purple). The SDS-PAGE of the elution fractions are shown beneath the profiles. Asterisks indicate fractions used for mass photometry and single molecule assays. The number shown at the left side indicates molecular weight standard. (**C**) Mass photometry of KLP-6 (1–587) and KLP-6 (1–587)(D458A). Histograms show the normalized particle count of KLP-6 (1–587) (blue) and KLP-6 (1–587)(D458A) (orange) at 40 nM. Lines show Gaussian fits (mean ± SD: 94±44 kDa and 88±35 kDa for KLP-6 (1–587) and KLP-6 (1–587)(D458A), respectively). (**D**) Representative kymographs showing the motility of 5 pM KLP-6 (1–587) and KLP-6 (1–587)(D458A) in the presence of 2 mM ATP. Note that no directional movement was detected in either case. Horizontal and vertical bars show 10 μm and 10 s, respectively. (**E**) Dot plots showing the landing rate of KLP-6 (1–587) and KLP-6 (1–587)(D458A). Each dot shows a single datum point. Green bars represent median value. n=33 microtubules. Mann-Whitney U test. ****, p<0.0001. (**F**) Dot plots showing the dwell time of KLP-6 (1–587) and KLP-6 (1–587)(D458A) on microtubules. Each dot shows a single datum point. Green bars represent median value and interquartile range. n=338 and 351 particles for KLP-6 (1–587) and KLP-6 (1–587)(D458A), respectively. Mann-Whitney U test. ****, p<0.0001.

The online version of this article includes the following source data and figure supplement(s) for figure 4:

**Source data 1.** Landing rate and dwell time of KLP-6 (1–587) and KLP-6 (1–587)(D458A).

**Source data 2.** Original file for the SDS-PAGE analysis in *Figure 4B* (KLP-6(1-587) and KLP-6(1-587)(D458A)).

**Source data 3.** PDF containing *Figure 4B* and original scans of the relevant SDS-PAGE analysis (KLP-6(1-587) and KLP-6(1-587)(D458A)) with highlighted bands.

**Figure supplement 1.** Comparison of KLP-6 and UNC-104.

*Figure 4 continued on next page*

*Figure 4 continued*

**Figure supplement 2.** Structural model of KLP-6 and KIF13B.

*et al., 2018*; *Wang et al., 2022*). Therefore, we introduced the E412K mutation into UNC-104 and analyzed biochemical properties (*Figure 5A*). UNC-104(1–653) exhibited two peaks in the SEC analysis (*Figure 5B*), corresponding to the size of dimers and monomers, respectively. UNC-104(1–653) predominantly eluted in the monomer peak (*Figure 5B*, lower panel). This monomer peak fraction was subjected to a re-run in the SEC analysis. After incubating for sufficient time, we observed that the monomeric fraction re-equilibrated into two peaks, resembling the results from the initial SEC analysis of UNC-104(1–653) (*Figure 5B* and *Figure 5—figure supplement 1*). Unlike KLP-6 (1–587), UNC-104(1–653) exhibited a clear peak shift in the SEC analysis by introducing a mutation to relieve the autoinhibition (*Figure 5B*). UNC-104(1–653)(E412K) eluted in a single peak whose expected molecular weight was that of a dimer. To confirm that the peak shift was due to the conversion from monomer to dimer, we analyzed the main peak fractions from UNC-104(1–653) and UNC-104(1–653) (E412K) by mass photometry. Mass photometry revealed that wild-type UNC-104(1–653) was mostly monomeric (*Figure 5C*, blue), with less than 5% of dimers detected. In contrast, UNC-104(1–653) (E412K) was a mixture of monomers and dimers (*Figure 5C*, orange), with the ratio of monomers and dimers almost 1:1. It is likely that the difference between SEC and mass photometry can be attributed to the concentration disparity between the two techniques; SEC operates in the micromolar range, while mass photometry operates in the nanomolar range, as further discussed later. The prior study showed that UNC-104(1–653) does not show any processive runs on *Chlamydomonas* axonemes, which were widely used in the single molecule assays (*Tomishige et al., 2002*). However, we found wild-type UNC-104(1–653) exhibited a trace number of processive runs on microtubules purified from brain tissues but not on *Chlamydomonas* axonemes (*Figure 5D* and *Figure 5—figure supplement 2*). Furthermore, UNC-104(1–653)(E412K) very frequently exhibited processive movement on purified microtubules (*Figure 5D–G*). Although the velocity of moving particles did not differ significantly (*Figure 5F*), the run length of UNC-104(1–653)(E412K) was much longer than that of wild-type UNC-104(1–653) (*Figure 5G*). These results suggest that the equilibrium between monomers and dimers is shifted toward the dimer form when the autoinhibition of UNC-104 is released. Finally, we confirmed that the difference between UNC-104 and KLP-6 is not due to the effect of different mutations because KLP-6(E409K) did not convert to dimers (*Figure 4—figure supplement 2* and *Figure 5—figure supplement 3*).

## The CC2 domain of UNC-104 is essential for the dimerization and processive motility

The sequences of KLP-6 (1–587) and UNC-104(1–653) are very similar, except that the CC2 domain of KLP-6 is much shorter than UNC-104 as depicted in *Figure 4—figure supplement 1*. We analyzed KLP-6 (1–587) and UNC-104(1–653) with a coiled-coil prediction algorism called Marcoil (*Delorenzi and Speed, 2002*), which is one of the most reliable prediction algorisms (*Gruber et al., 2006*). Although the X-ray structure of monomeric KLP-6 has suggested a short α-helix domain is the CC2 domain of KLP-6 (*Wang et al., 2022*), the probability of coiled-coil formation expected from the Marcoil algorism is very low (*Figure 6A*). On the other hand, the CC2 domain of UNC-104 is expected to be a conventional coiled coil with a high probability (*Figure 6A*). This difference in the structure may contribute to the distinct biochemical properties observed between KLP-6 and UNC-104. To compare the properties of the CC2 domains of KLP-6 and UNC-104, we fused them with a fluorescent protein mScarlet (*Figure 6B*). We analyzed the purified proteins by SEC (*Figure 6C*). While the peak of KLP-6CC2-mScarlet was indistinguishable from mScarlet alone, UNC-104CC2-mScarlet exhibited a clear peak shift (*Figure 6C*). The analysis indicates that UNC-104CC2, but not KLP-6CC2, is capable to induce dimerization.

To further explore the importance of the CC2 domain in the motility of UNC-104, we examined UNC-104(1–594) which consists of the MD, NC, CC1, and FHA domains, but not CC2 (*Figure 7A*). In the SEC analysis, both UNC-104(1–594) and UNC-104(1–594)(E412K) showed almost identical profiles (*Figure 7B*), which totally differed from the results that UNC-104(1–653)(E412K) shifted toward higher molecular weights compared to UNC-104(1–653) which presents two elution peaks (*Figure 5B*). Mass photometry showed that both UNC-104(1–594) and UNC-104(1–594) (E412K) were predominantly

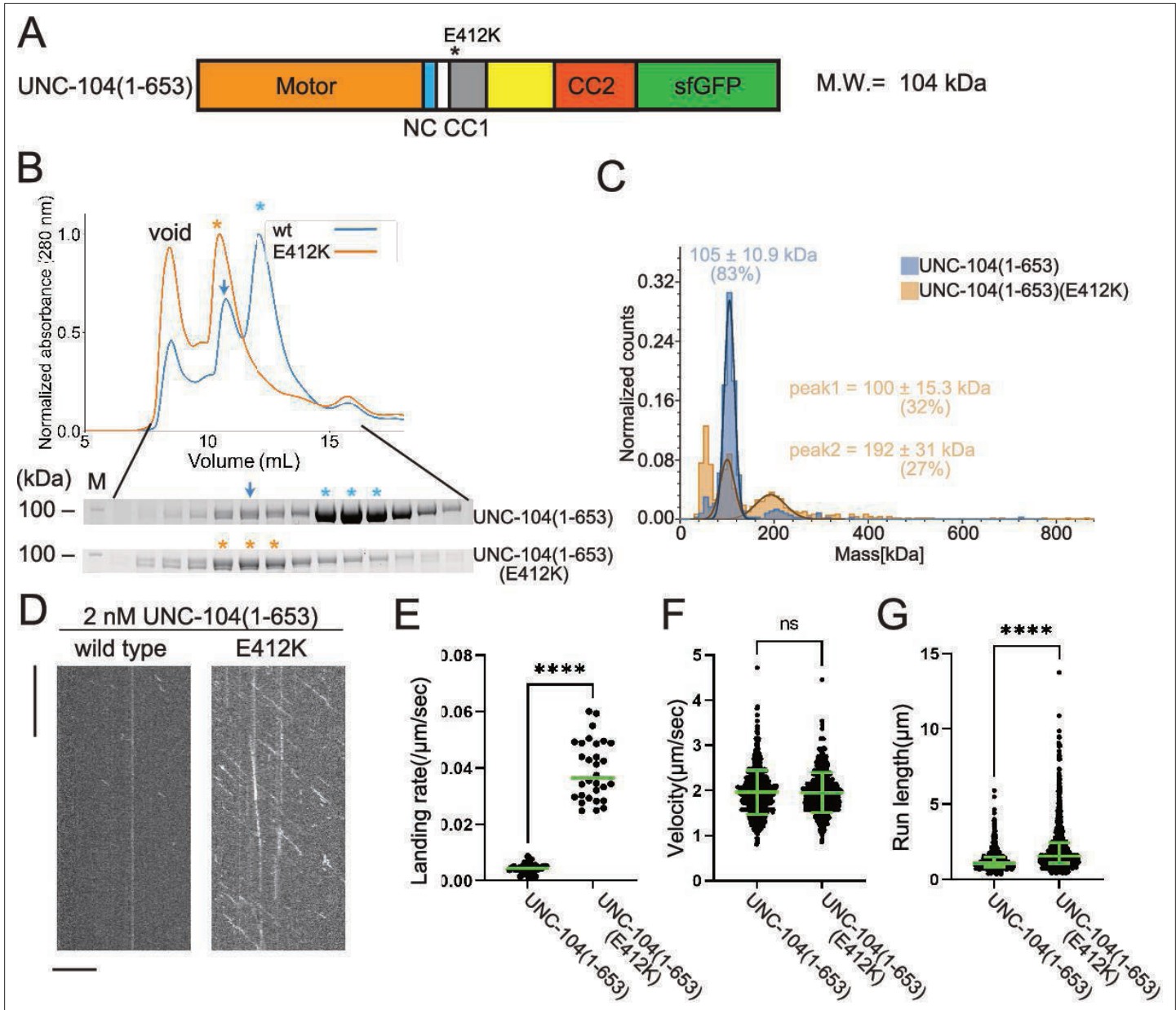

**Figure 5.** UNC-104 is converted to a processive dimer upon autoinhibition release. (**A**) Schematic drawing of the domain organization of UNC-104(1–653). Calculated molecular weight is written at the right side. (**B**) Size exclusion chromatography of UNC-104(1–653) (blue) and UNC-104(1–653)(E412K) (orange). The SDS-PAGE of the elution fractions are shown beneath the profiles. Blue and orange asterisks indicate fractions used for mass photometry and single molecule assays. The number shown at the left side indicates molecular weight standard. Blue arrows indicate expected dimer fraction of wild-type UNC-104(1–653). (**C**) Mass photometry of UNC-104(1–653) and UNC-104(1–653)(E412K). Histograms show the normalized particle count of UNC-104(1–653) (blue) and UNC-104(1–653)(E412K) (orange) at 10 nM. Lines show Gaussian fits. UNC-104(1–653) is distributed within 100±15.3 kDa range (mean ± SD). UNC-104(1–653)(E412K) shows two peaks consisting of 32% and 27% of particles which are distributed within 100±15.3 kDa and 192±31 kDa range, respectively (mean ± SD). Note that wild-type UNC-104(1–653) also has a very small peak around 200 kDa, but the number of datum point is too little for Gaussian fitting. (**D**) Representative kymographs showing the motility of 2 nM UNC-104(1–653) and UNC-104(1–653)(E412K) in the presence of 2 mM ATP. Horizontal and vertical bars show 10 μm and 10 s, respectively. (**E**) Dot plots showing the landing rate of UNC-104(1–653) and UNC-104(1–653)(E412K). Each dot shows a single datum point. Green bars represent median value n=31 and 30 microtubules for UNC-104(1–653) and UNC-104(1–653)(E412K), respectively. Mann-Whitney U test. ****, p<0.0001. (**F**) Dot plots showing the velocity of UNC-104(1–653) and UNC-104(1–653) (E412K). Each dot shows a single datum point. Green bars represent mean ± SD, n=603 and 624 particles for UNC-104(1–653) and UNC-104(1–653) (E412K). Student's t-test, ns, p=0.7 and statistically not significant. (**G**) Dot plots showing the run length of UNC-104(1–653) and UNC-104(1–653)(E412K). Each dot shows a single datum point. Green bars represent median value and interquartile range. n=603 and 624 particles for UNC-104(1–653) and UNC-104(1–653)(E412K). Mann-Whitney U test. ****, p<0.0001.

*Figure 5 continued on next page*

*Figure 5 continued*

The online version of this article includes the following source data and figure supplement(s) for figure 5:

**Source data 1.** Landing rate, run length, and velocity of UNC-104(1–653) and UNC-104(1–653)(E412K).

**Source data 2.** Original file for the SDS-PAGE analysis in *Figure 5B* (UNC-104(1-653) and UNC-104(1-653)(E412K)).

**Source data 3.** PDF containing *Figure 5B* and original scans of the relevant SDS-PAGE analysis (UNC-104(1-653) and UNC-104(1-653)(E412K)) with highlighted bands.

**Figure supplement 1.** Monomer-dimer re-equilibrium of monomeric UNC-104(1–653).

**Figure supplement 1—source data 1.** Original file for the SDS-PAGE analysis in *Figure 5—figure supplement 1* (UNC-104(1-653)).

**Figure supplement 1—source data 2.** PDF containing Figure 5-figure supplement 1 and original scans of the relevant SDS-PAGE analysis (UNC-104(1-653)) with highlighted bands.

**Figure supplement 2.** Single molecule analysis using *Chlamydomonas* axonemes.

**Figure supplement 2—source data 1.** Landing rate of UNC-104(1–653) on microtubules and axonemes.

**Figure supplement 3.** KLP-6(E409K) does not form a dimer.

**Figure supplement 3—source data 1.** Original file for the SDS-PAGE analysis in *Figure 5—figure supplement 3* (KLP-6(1-587)(E409K)).

**Figure supplement 3—source data 2.** PDF containing *Figure 5—figure supplement 3* and a original scan of the relevant SDS-PAGE analysis (KLP-6(1-587)(E409K)) with a highlighted band.

monomeric in solution (*Figure 7C*). TIRF microscopy analysis demonstrated that, UNC-104(1–594) rarely showed processive runs on microtubules at the nanomolar range. While UNC-104(1–594)(E412K) displayed some processive runs on microtubules, the run length and landing rate were much lower than those of UNC-104(1–653)(E412K) (*Figure 7D–G*). These results suggest that the CC2 domain is required for the stable dimer formation, activation, and processive runs of UNC-104. To confirm the importance of CC2-dependent dimerization in the axonal transport, we proceeded to biochemically analyze a mutation within the CC2 domain of UNC-104. The mutation, UNC-104(L640F), reduces the axonal transport of synaptic materials in *C. elegans* (*Cong et al., 2021*). This mutation was introduced into UNC-104 (1–653) (E412K). SEC analysis showed that UNC-104 (1–653) (E412K, L640F) predominantly eluted in the monomer peak, even though E412K mitigated autoinhibition and promoted dimerization (*Figure 7—figure supplement 1*). This in vitro result is consistent with abnormal accumulation of synaptic vesicles in *unc-104(L640F)* worms.

## Discussion

### Equilibrium between monomers and dimers in UNC-104

Our SEC analysis showed that UNC-104(1–653) was a mixture of dimers and monomers, with monomers predominating (*Figure 5B*). A prior study has observed similar phenomena. UNC-104(1–653) is able to form active dimers at micromolar concentrations (*Tomishige et al., 2002*). In contrast, UNC-104(1–653)(E412K) showed a different profile in the SEC analysis, with the majority of the protein recovered from dimer fractions (*Figure 5B*). However in the mass photometry, UNC-104(1–653)(E412K) was a 1:1 mixture of dimers and monomers (*Figure 5C*). This difference could be due to the different concentrations used in the two techniques, with the SEC analysis performed at micromolar concentrations and mass photometry at nanomolar concentrations. These results suggest that (1) UNC-104 exists in an equilibrium between monomers and dimers and (2) the release of the autoinhibition, such as by UNC-104(E412K) mutation, shifts the state of the equilibrium toward the formation of dimers. In contrast, we have not detected KLP-6 dimers in either SEC or mass photometry.

### The CC2 domain is required for the processive movement

Our results indicate that the CC2 domain is essential for the formation of stable dimers (*Figure 7C*). The CC2 domain has been demonstrated to be crucial for the autoinhibition of KIF1A and UNC-104 (*Hammond et al., 2009*; *Lee et al., 2004*). We have previously shown that the E612K mutation in the CC2 domain of UNC-104 disrupt its autoinhibition (*Niwa et al., 2016*). Moreover, crystal structure analysis revealed that the short CC2 domain of KLP-6 is also involved in autoinhibition. In addition to the role in the autoinhibition, CC2 domain is required for the cargo binding (*Hummel and Hoogenraad, 2021*). We suggest here that, in addition to these functions, CC2 is required for the

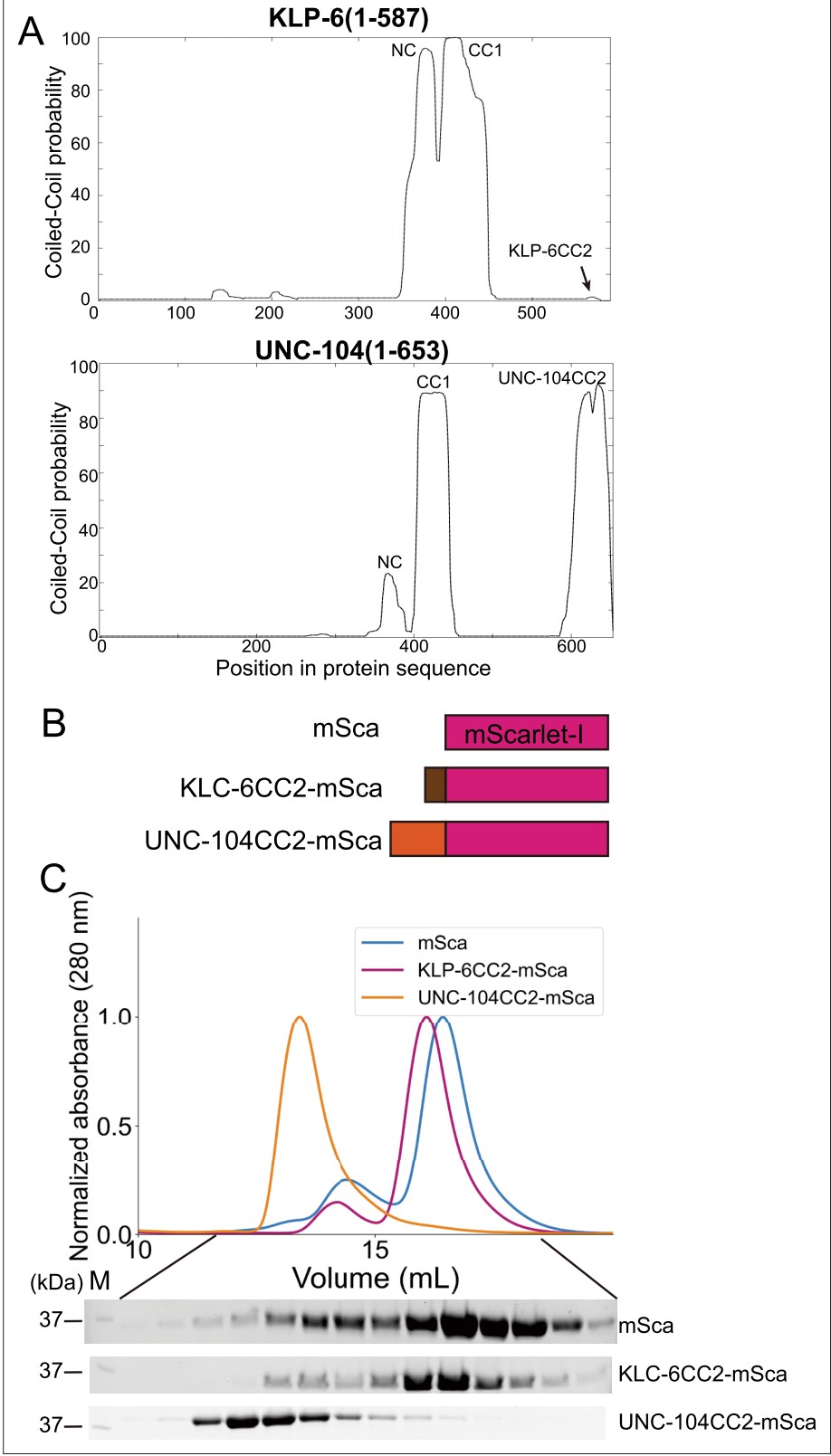

**Figure 6.** Coiled-coil 2 (CC2) domain of UNC-104, but not KLP-6, is capable to form a dimer. (**A**) Coiled coil prediction of KLP-6 (aa 1–587) and UNC-104 (aa 1–653). NC, CC1, and CC2 domains are indicated. Prediction was performed using the Marcoil algorism. (**B**) Schematic drawing of the domain organization of mScarlet-I(mSca), KLP-6CC2-mSca, and UNC-104CC2-mSca analyzed in panel C. (**C**) Size exclusion chromatography of mSca (blue), KLP-

*Figure 6 continued on next page*

*Figure 6 continued*

6CC2-mSca (plum), and UNC-104CC2-mSca (orange). The SDS-PAGE of the elution fractions are shown beneath the profiles. The number shown at the left side indicates molecular weight standard.

The online version of this article includes the following source data for figure 6:

**Source data 1.** Original file for the SDS-PAGE analysis in *Figure 6C* (mSca, KLP-6CC2-mSca, and UNC-104CC2-mSca).

**Source data 2.** PDF containing *Figure 6C* and original scans of the relevant SDS-PAGE analysis (mSca, KLP-6CC2-mSca, and UNC-104CC2-mSca) with highlighted bands.

full activation and processivity of kinesin-3. Because KLP-6 does not have a conserved CC2 domain, KLP-6 cannot form stable dimers (*Figure 4B and C*), although KLP-6 needs to be dimerized to obtain processivity (*Figure 3B–E*). A prior work has shown that a deletion mutant of KIF13B composed of MD, NC, and CC1 can form a dimer when the autoinhibition is unlocked by mutations, but the major population remains monomeric in the ultracentrifugation analysis (*Ren et al., 2018*). This would be because CC2 is not included in their assays. Indeed, the monomer-to-dimer conversion was observed upon release of the autoinhibition in the case of the full length of KIF13B which has the CC2 domain (*Fan and McKenney, 2022*). The CC2 domain is conserved in most of the kinesin-3 family proteins including KIF1Bα, KIF1C, KIF13A, KIF16A, and KIF16B (*Hirokawa et al., 2009*). Taken together, we suggest that the CC2 domain is fundamental to the stable dimer formation in kinesin-3. Moreover, our data suggest that the CC2 domain is required for UNC-104 to move long distances on microtubules (*Figure 7D–G*). Without the CC2 domain, UNC-104 shows only very short runs even if they are activated for microtubule binding. Therefore, the CC2 domain would be essential for long-distance transport, such as axonal transport. Consistent with this idea, the CC2 mutation, unc-104(L640F), which inhibits efficient dimerization of UNC-104 in vitro (*Figure 7—figure supplement 1*), reduces the amount of axonal transport in vivo (*Cong et al., 2021*).

## Activation of UNC-104 and KLP-6

Neither Tomishige et al. nor we were able to observe any processive runs of wild-type UNC-104(1-653) on *Chlamydomonas* axonemes (*Figure 5—figure supplement 2*; *Tomishige et al., 2002*). Moreover, the motility of KIF1A(1-393)LZ was less frequent on *Chlamydomonas* axonemes compared with microtubules purified from brain tissues (*Figure 5—figure supplement 2*). These observation might be attributed to the extensive post-translational modifications and the presence of microtubule binding proteins on *Chlamydomonas* axonemes, which potentially inhibits the motility of kinesins. In contrast, we show that processive movements of UNC-104 can be observed even at nanomolar concentrations when employing purified porcine microtubules as the tracks. Moreover, our study provides compelling evidence that the release of autoinhibition is sufficient for monomeric UNC-104 to dimerize and move processively on microtubules at low nanomolar concentrations. These results revisit previous notion that UNC-104 needs to be enriched to PIP2 microdomains on cargo vesicles to dimerize and move processively (*Figure 7—figure supplement 2A*; *Klopfenstein et al., 2002*; *Tomishige et al., 2002*). A recent study has shown that KIF1A, a mammalian ortholog of UNC-104, needs to form a homodimer to bind to cargo vesicles, indicating that KIF1A dimerizes before interacting with cargo vesicles, rather than on cargo vesicles (*Hummel and Hoogenraad, 2021*). Based on these findings, we propose that UNC-104 forms dimers in cytosol upon release of autoinhibition, rather than on cargo vesicles, as a prerequisite for motor activation and cargo binding (*Figure 7—figure supplement 2B*). Indeed, the amount of synaptic vesicles transported by UNC-104 is increased by the UNC-104(E412K) mutation (*Niwa et al., 2016*). It is possible that PIP2 microdomains may play a role in stabilizing UNC-104 dimers to ensure long-distance axonal transport because UNC-104 dimers are more stable at higher concentrations even when the autoinhibition is unlocked (*Figure 5B and C*).

Unlike UNC-104, autoinhibition release is insufficient to induce dimerization of KLP-6. KLP-6 is monomeric in solution even when autoinhibition is unlocked (*Figure 4*). LOV-1 and PKD-2, which form the mechanoreceptor complex transported by KLP-6, stand as potential candidates for an additional regulatory role in KLP-6 dimerization. However, KLP-6 can be activated in Neuro 2a cells when autoinhibition is unlocked (*Wang et al., 2022*), implying that other unidentified factors, including post-translational modifications and binding proteins, may be required for KLP-6 dimerization

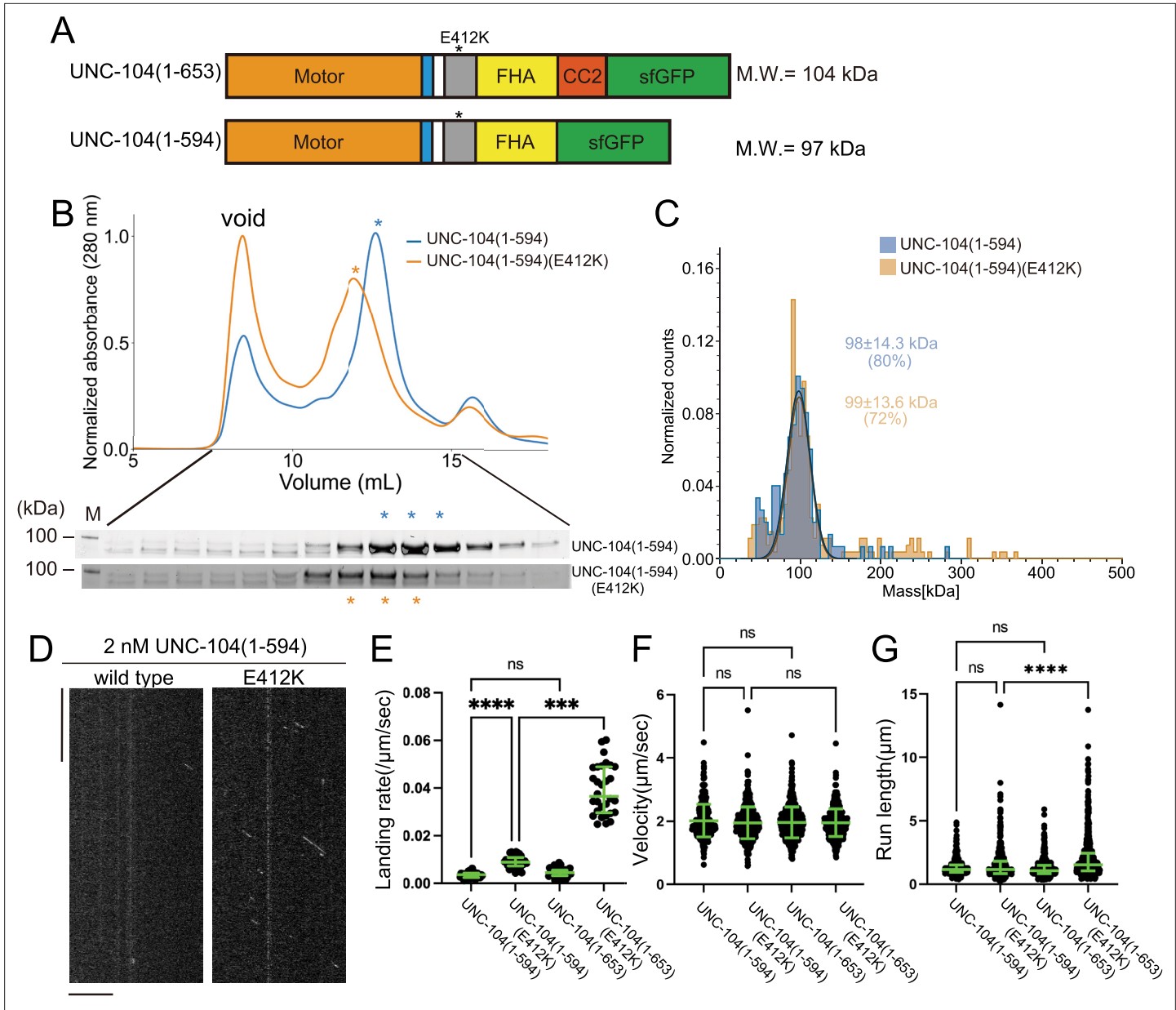

**Figure 7.** The coiled-coil 2 (CC2) domain is essential for the stable dimer formation in UNC-104. (**A**) Schematic drawing of the domain organization of UNC-104(1–653) and UNC-104(1–594). Calculated molecular weight is shown at the right side. (**B**) Size exclusion chromatography of UNC-104(1–594) (blue) and UNC-104(1–594)(E412K) (orange). The SDS-PAGE of the elution fractions are shown beneath the profiles. The number shown at the left side indicates molecular weight standard. Both proteins show almost identical profile. (**C**) Mass photometry of UNC-104(1–594) (blue) and UNC-104(1–594) (E412K) (orange) at 10 nM. Lines show Gaussian fits. Both UNC-104(1–594) and UNC-104(1–594)(E412K) have a single peak which is Gaussian distributed within 98±14.3 kDa range and within 99±13.6 kDa range, respectively (mean ± SD). (**D**) Representative kymographs showing the motility of 2 nM UNC-104(1–594) and UNC-104(1–594)(E412K) in the presence of 2 mM ATP. Horizontal and vertical bars show 10 μm and 10 s, respectively. (**E**) Dot plots showing the landing rate of UNC-104(1–594), UNC-104(1–594)(E412K), UNC-104(1–653), and UNC-104(1–653)(E412K). For comparison, data for UNC-104(1–653) and UNC-104(1–653)(E412K) are replotted from *Figure 5E*. Each dot shows a single datum point. Green bars represent median value and interquartile range. N=31, 32, 31, 30 microtubules for UNC-104(1–594), UNC-104(1–594)(E412K), UNC-104(1–653), and UNC-104(1–653)(E412K), respectively. Kruskal-Wallis test followed by Dunn's multiple comparison test. ***, p<0.001. ****, p<0.0001. ns, p>0.05 and statistically not significant. (**F**) Dot plots showing the velocity of UNC-104(1–594), UNC-104(1–594)(E412K), UNC-104(1–653), and UNC-104(1–653)(E412K). For comparison, data for UNC-104(1–653) and UNC-104(1–653)(E412K) are replotted from *Figure 5F*. Green bars represent mean ± SD, n=467, 609, 603, 624 particles for UNC-104(1–594), UNC-104(1–594)(E412K), UNC-104(1–653), and UNC-104(1–653)(E412K), respectively. One-way ANOVA test followed by Šidák's multiple comparison test. ns, p>0.05 and statistically not significant. (**G**) Dot plots showing the run length of UNC-104(1–594), UNC-104(1–594)(E412K), UNC-104(1–653), and UNC-104(1–653)(E412K). For comparison, data for UNC-104(1–653) and UNC-104(1–653)(E412K) are replotted from *Figure 5G*. Each dot

*Figure 7 continued on next page*

*Figure 7 continued*

shows a single datum point. Green bars represent median value and interquartile range. Kruskal-Wallis test followed by Dunn's multiple comparison test. n=467, 609, 603, 624 particles for UNC-104(1–594), UNC-104(1–594)(E412K), UNC-104(1–653), and UNC-104(1–653)(E412K), respectively. ****, p<0.0001. ns, p>0.05 and statistically not significant.

The online version of this article includes the following source data and figure supplement(s) for figure 7:

**Source data 1.** Landing rate, run length, and velocity of UNC-104(1–594) and UNC-104(1–594)(E412K).

**Source data 2.** Original file for the SDS-PAGE analysis in *Figure 7B* (UNC-104(1-594) and UNC-104(1-594)(E412K)).

**Source data 3.** PDF containing *Figure 7B* and original scans of the relevant SDS-PAGE analysis (UNC-104(1-594) and UNC-104(1-594)(E412K)) with highlighted bands.

**Figure supplement 1.** UNC-104(E412K,L640F) does not form a dimer.

**Figure supplement 1—source data 1.** Original file for the SDS-PAGE analysis in *Figure 7—figure supplement 1* (UNC-104(1-653)(E412K,L640F)).

**Figure supplement 1—source data 2.** PDF containing *Figure 7—figure supplement 1* and a original scan of the relevant SDS-PAGE analysis (UNC-104(1-653)(E412K,L640F)) with a highlighted band.

**Figure supplement 2.** Activation models for UNC-104 and KLP-6.

**Figure supplement 3.** Proteins analyzed in this study.

---

(*Figure 7—figure supplement 2C*). Our findings imply that mutations disrupting autoinhibition of KLP-6, such as D458A mutation, enhance the intraflagellar transport of LOV-1 and PKD-2 complex in *C. elegans*. Further investigations are needed to fully unravel the molecular mechanisms underlying the dimerization and activation of these motors by identifying factors that can unlock the autoinhibition of UNC-104 and KLP-6 in vitro, as well as factors that induce the dimerization of KLP-6.

## Methods

### Preparation of klp-6 and unc-104 cDNA

*Wild-type* N2 strain was obtained from *C. elegans* genetic center (MN, USA). Total RNA was purified from N2 using Nucleospin RNA (Takara Bio Inc, Kusatsu, Japan) as described in the manufacturer's procedure. From the total RNA, cDNA was obtained using Superscript IV reverse transcriptase in combination with Oligo dT primer (Thermo Fisher Scientific Japan, Tokyo, Japan). *klp-6* cDNA was amplified by polymerase chain reaction (PCR) using KOD FX neo DNA polymerase (TOYOBO, Tokyo, Japan). Following primers were used: *klp-6_F*, ATGGGAAAGGGTGACTCCATAATCG *klp-6_R*, CCCT TATTTCGCCTTTGGTTTCTTCG *unc-104* cDNA was codon optimized for *Spodoptera frugiperda* and synthesized by Geneart (Thermo Fisher Scientific) because we could not express enough amount of UNC-104 using the original *unc-104* cDNA. *klp-6* and *unc-104* cDNA fragments were amplified by PCR and cloned into the pAcebac1-sfGFP vector (*Chiba et al., 2019*). KLP-6 (1–390)::LZ::mScarlet-I::Strep-tag II was generated by Gibson assembly (*Gibson et al., 2009*) based on the KIF1A(1-393)::LZ::mScarlet-I::Strep-tag II vector (*Anazawa et al., 2022*).

### Expression of KLP-6 and UNC-104 in Sf9 cells

Sf9 cells (Thermo Fisher Scientific) were maintained in Sf900 II SFM (Thermo Fisher Scientific) at 27°C. DH10Bac (Thermo Fisher Scientific) were transformed to generate bacmid. To prepare baculovirus, $1\times10^6$ cells of Sf9 cells were transferred to each well of a tissue culture-treated six-well plate. After the cells attached to the bottom of the dishes, about ~5 µg of bacmid were transfected using 5 µL of TransIT-Insect transfection reagent (Takara Bio Inc). Five days after initial transfection, the culture media were collected and spun at 3000×*g* for 3 min to obtain the supernatant (P1). For protein expression, 400 mL of Sf9 cells ($2\times10^6$ cells/mL) were infected with 200 µL of P1 virus and cultured for 65 hr at 27°C. Cells were harvested and stocked at –80°C.

### Expression of KLP-6 in *Escherichia coli*

To express KLP-6 (1–390)::LZ::mScarlet-I::Strep-tag II, LOBSTR(DE3) (*Andersen et al., 2013*) was transformed and selected on an LB agar plate supplemented with kanamycin at 37°C. Colonies were picked and cultured in 10 mL LB medium supplemented with kanamycin overnight. Next morning, 5 mL of the medium was transferred to 500 mL 2.5×YT (20 g/L tryptone, 12.5 g/L yeast

extract, 6.5 g/L NaCl) supplemented with 10 mM phosphate buffer (pH 7.4) and 50 µg/mL kanamycin in a 2 L flask and shaken at 37°C. When $OD_{600}$ reached 0.6, flasks were cooled in ice-cold water for 30 min. Then, 23.8 mg IPTG was added to each flask. Final concentration of IPTG was 0.2 mM. Flasks were shaken at 18°C overnight. Next day, bacteria expressing recombinant proteins were pelleted by centrifugation (3000×$g$, 10 min, 4°C), resuspended in PBS and centrifuged again (3000×$g$, 10 min, 4°C). Pellets were resuspended in protein buffer (50 mM HEPES-KOH, pH 8.0, 150 mM $KCH_3COO$, 2 mM $MgSO_4$, 1 mM EGTA, 10% glycerol) supplemented with phenylmethylsulfonyl fluoride (PMSF).

## Purification of recombinant proteins

Sf9 cells were resuspended in 25 mL of lysis buffer (50 mM HEPES-KOH, pH 7.5, 150 mM $KCH_3COO$, 2 mM $MgSO_4$, 1 mM EGTA, 10% glycerol) along with 1 mM DTT, 1 mM PMSF, 0.1 mM ATP, and 0.5% Triton X-100. After incubating on ice for 10 min, lysates were cleared by centrifugation (15,000×$g$, 20 min, 4°C) and subjected to affinity chromatography described below.

Bacteria were lysed using a French Press G-M (Glen Mills, NJ, USA) as described by the manufacturer. After being incubated with 1% streptomycin sulfate on ice for 20 min to eliminate nucleic acids from protein samples (*Liang et al., 2009*), lysates were cleared by centrifugation (75,000×$g$, 20 min, 4°C) and subjected to affinity chromatography described below.

Lysate was loaded on Streptactin-XT resin (IBA Lifesciences, Göttingen, Germany) (bead volume: 2 mL). The resin was washed with 40 mL Strep wash buffer (50 mM HEPES-KOH, pH 8.0, 450 mM $KCH_3COO$, 2 mM $MgSO_4$, 1 mM EGTA, 10% glycerol). Protein was eluted with 40 mL Strep elution buffer (50 mM HEPES-KOH, pH 8.0, 150 mM $KCH_3COO$, 2 mM $MgSO_4$, 1 mM EGTA, 10% glycerol, 300 mM biotin). Eluted solution was concentrated using an Amicon Ultra 15 (Merck) and then separated on an NGC chromatography system (Bio-Rad) equipped with a Superdex 200 Increase 10/300 GL column (Cytiva). Peak fractions were collected and concentrated using an Amicon Ultra 4 (Merck). Proteins were analyzed by SDS-PAGE followed by CBB staining (*Figure 7—figure supplement 3*). Concentrated proteins were aliquoted and snap-frozen in liquid nitrogen.

## Mass photometry

Proteins obtained from the peak fractions in the SEC analysis were pooled, snap-frozen, and stored until measurement. Prior to measurement, the proteins were thawed and diluted to a final concentration 5–10 nM in protein buffer without glycerol. Mass photometry was performed using a Refeyn OneMP mass photometer (Refeyn Ltd, Oxford, UK) and Refeyn AcquireMP version 2.3 software, with default parameters set by Refeyn AcquireMP. Bovine serum albumin (BSA) was used as a control to determine the molecular weight. The results were subsequently analyzed using Refeyn DiscoverMP version 2.3, and graphs were prepared to visualize the data.

## Preparation of microtubules and axonemes

Tubulin was purified from porcine brain as described (*Castoldi and Popov, 2003*). Tubulin was labeled with Biotin-PEG$_2$-NHS ester (Tokyo Chemical Industry, Tokyo, Japan) and AZDye647 NHS ester (Fluoroprobes, Scottsdale, AZ, USA) as described (*Al-Bassam, 2014*). To polymerize Taxol-stabilized microtubules labeled with biotin and AZDye647, 30 µM unlabeled tubulin, 1.5 µM biotin-labeled tubulin, and 1.5 µM AZDye647-labeled tubulin were mixed in BRB80 buffer supplemented with 1 mM GTP and incubated for 15 min at 37°C. Then, an equal amount of BRB80 supplemented with 40 µM taxol was added and further incubated for more than 15 min. The solution was loaded on BRB80 supplemented with 300 mM sucrose and 20 µM taxol and ultracentrifuged at 100,000×$g$ for 5 min at 30°C. The pellet was resuspended in BRB80 supplemented with 20 µM taxol.

*Chlamydomonas* axonemes were prepared as described (*Hou et al., 2021*). Flagella were de-membranated by resuspension in HMDEK (30 mM HEPES, 5 mM $MgSO_4$, 1 mM DTT, 0.5 mM EGTA, 25 mM KCl) containing 0.1% NP-40 for 10 min at 4°C. The solution was centrifuged at 20,000×$g$ for 10 min at 4°C. The pellet was resuspended in HMDEK.

## TIRF single molecule motility assays

TIRF assays using the purified porcine microtubules were performed as described (*Chiba et al., 2019*). Glass chambers were prepared by acid washing as previously described (*Chiba et al., 2022*). Glass chambers were coated with PLL-PEG-biotin (SuSoS, Dübendorf, Switzerland). Polymerized

microtubules were flowed into streptavidin adsorbed flow chambers and allowed to adhere for 5–10 min. Unbound microtubules were washed away using assay buffer (90 mM HEPES-KOH pH 7.4, 50 mM KCH$_3$COO, 2 mM Mg(CH$_3$COO)$_2$, 1 mM EGTA, 10% glycerol, 0.1 mg/mL biotin-BSA, 0.2 mg/mL kappa-casein, 0.5% Pluronic F127, 2 mM ATP, and an oxygen scavenging system composed of PCA/PCD/Trolox). Purified motor protein was diluted to indicated concentrations in the assay buffer. Then, the solution was flowed into the glass chamber.

For TIRF assays using *Chlamydomonas* axonemes, axonemes were first flowed into the empty grass chambers, and then PLL-PEG-biotin (SuSoS) was flowed into the glass chamber. Unbound axonemes and PLL-PEG-biotin were washed away using assay buffer. UNC-104(1-653)::sfGFP::Strep-tag II was diluted to 10 nM in the assay buffer. Furthermore, KIF1A(1-393)::LZ::mScarlet-I::Strep-tag II (*Anazawa et al., 2022*), a constitutively active motor, was diluted to 0.2 nM in the same buffer for the purpose of labeling and visualizing axonemes. The solution was flowed into the glass chamber.

An ECLIPSE Ti2-E microscope equipped with a CFI Apochromat TIRF 100XC Oil objective lens (1.49 NA), an Andor iXion life 897 camera and a Ti2-LAPP illumination system (Nikon, Tokyo, Japan), was used to observe single molecule motility. NIS-Elements AR software version 5.2 (Nikon) was used to control the system. At least three independent experiments were conducted for each measurement.

## Microtubule gliding assays

Tubulin was labeled with AZDye647, and the labeled microtubules were prepared without biotin-labeled tubulin as described in *Preparation of microtubules and axonemes*. Microtubule gliding assays were performed using two distinct methods that varied in the way the motors were attached to the glass surface. The first method was that the motors diluted in BRB80 buffer were first flowed into the empty grass chambers and attached to the glass surface directly. The second method was described as follows. Glass chambers were coated with PLL-PEG-biotin (SuSoS). Streptavidin is adsorbed in flow chambers. Biotin-labeled anti-GFP antibodies (MBL Life Science, Tokyo, Japan) diluted in BRB80 buffer were flowed into the chamber. Chamber was washed by assay buffer. sfGFP-tagged motors were diluted in assay buffer and flowed into the chamber. Chamber was washed by assay buffer again. Microtubules diluted by assay buffer was flowed into the chamber and analyzed by the ECLIPSE Ti2-E microscope equipped with a CFI Apochromat TIRF 100× oil objective lens (1.49 NA), an Andor iXion life 897 camera and a Ti2-LAPP illumination system (Nikon, Tokyo, Japan). At least three independent experiments were conducted for each measurement.

## Statistical analyses and graph preparation

Statistical analyses were performed using GraphPad Prism version 9. Statistical methods are described in the figure legends. The structure figures were prepared with PyMOL and the atomic coordinates were downloaded from the Protein Data Bank. Graphs were prepared using GraphPad Prism version 9, exported in the pdf format and aligned by Adobe Illustrator 2021.

# Acknowledgements

We thank Dr. T Kubo for providing the Chlamydomonas flagella. We also would like to thank the members of Niwa lab for useful discussions. SN was supported by JSPS KAKENHI (grants nos. 23H02472, 22H05523 and 20H03247). KC was supported by JSPS KAKENHI (grant no. 22K15053), Uehara Memorial Foundation, Naito Foundation, and MEXT Leading Initiative for Excellent Researchers (grant no. JPMXS0320200156). This work was performed under the Collaborative Research Program of Institute for Protein Research, Osaka University, CR-22-02.

# Additional information

### Funding

| Funder | Grant reference number | Author |
| --- | --- | --- |
| Japan Society for the Promotion of Science | 22H05523 | Shinsuke Niwa |

| Funder | Grant reference number | Author |
|--------|------------------------|--------|
| Japan Society for the Promotion of Science | 23H02472 | Shinsuke Niwa |
| Japan Society for the Promotion of Science | 20H03247 | Shinsuke Niwa |
| Japan Society for the Promotion of Science | 22K15053 | Kyoko Chiba |
| Uehara Memorial Foundation | | Kyoko Chiba |
| Naito Foundation | | Kyoko Chiba |
| Ministry of Education, Culture, Sports, Science and Technology | JPMXS0320200156 | Kyoko Chiba |

The funders had no role in study design, data collection and interpretation, or the decision to submit the work for publication.

## Author contributions

Tomoki Kita, Investigation, Visualization, Writing - original draft, Writing - review and editing; Kyoko Chiba, Shinsuke Niwa, Funding acquisition, Investigation, Visualization, Writing - original draft, Writing - review and editing; Jiye Wang, Investigation; Atsushi Nakagawa, Investigation, Writing - review and editing

## Author ORCIDs

Tomoki Kita http://orcid.org/0000-0002-1531-565X
Atsushi Nakagawa http://orcid.org/0000-0002-1700-7861
Shinsuke Niwa http://orcid.org/0000-0002-8367-9228

Reviewer #1 (Public Review): https://doi.org/10.7554/eLife.89040.3.sa1
Reviewer #2 (Public Review): https://doi.org/10.7554/eLife.89040.3.sa2
Reviewer #3 (Public Review): https://doi.org/10.7554/eLife.89040.3.sa3
Author Response https://doi.org/10.7554/eLife.89040.3.sa4

# Additional files

## Supplementary files

- MDAR checklist

## Data availability

All data generated or analysed during this study are included in the manuscript and supporting file. Source Data files have been provided for *Figures 1, 3–7*, *Figure 1—figure supplement 1*, *Figure 2—figure supplement 1*, *Figure 5—figure supplements 1–3*, *Figure 7—figure supplement 1*.

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
