## [Editor Report · eLife assessment]

This study explores the activation mechanisms of members of the kinesin-3 family, demonstrating common and unique regulation modes with **solid** evidence. The findings make for **valuable** contributions to the field of kinesin activation and regulation.

---

## [Referee Report · Reviewer #1 (Public Review)]

The regulation of motor autoinhibition and activation is essential for efficient intracellular transport. This manuscript used biochemical approaches to explore two members in the kinesin-3 family. They found that releasing UNC-104 autoinhibition triggered its dimerization whereas unlocking KLP-6 autoinhibition is insufficient to activate its processive movement, which suggests that KLP-6 requires additional factors for activation, highlighting the common and diverse mechanisms underlying motor activation. They also identified a coiled-coil domain crucial for the dimerization and processive movement of UNC-104. Overall, these biochemical and single-molecule assays were well performed, and their data support their statements. The manuscript is also clearly written, and these results will be valuable to the field.

---

## [Referee Report · Reviewer #2 (Public Review)]

The Kinesin superfamily motors mediate the transport of a wide variety of cargos which are crucial for cells to develop into unique shapes and polarities. Kinesin-3 subfamily motors are among the most conserved and critical classes of kinesin motors which were shown to be self-inhibited in a monomeric state and dimerize to activate motility along microtubules. Recent studies have shown that different members of this family are uniquely activated by to undergo transition from monomers to dimers.

Niwa and colleagues study two well-described members of the kinesin-3 superfamily, unc104 and KLP6, to uncover the mechanism of monomer to dimer transition upon activation. Their studies reveal that although both Unc104 and KLP6 are both self-inhibited monomers, their propensities for forming dimers are quite different. The authors relate this difference to a region in the molecules called CC2 which has a higher propensity for forming homodimers. Unc104 readily forms homodimers if its self-inhibited state is disabled while KLP6 does not.

The work suggests that although mechanisms for self-inhibited monomeric states are similar, variations in the kinesin-3 dimerization may present a unique forms of kinesin-3 motor regulation with implications on the forms of motility functions carried out by these unique kinesin-3 motors.

---

## [Referee Report · Reviewer #3 (Public Review)]

In this work, Kita et al., aim to understand the activation mechanisms of the kinesin-3 motors KLP-6 and UNC-104 from *C. elegans*. As many other motor proteins involved in intracellular transport processes, KLP-6 and UNC-104 motors suppress their ATPase activities in the absence of cargo molecules. Relieving the autoinhibition is thus a crucial step that initiates directional transport of intracellular cargo. To investigate the activation mechanisms, the authors make use of mass photometry to determine the oligomeric states of the full length KLP-6 and the truncated UNC-104(1-653) motors at sub-micromolar concentrations. While full length KLP-6 remains monomeric, the truncated UNC-104(1-653) displays a sub-population of dimeric motors that is much more pronounced at high concentrations, suggesting a monomer-to-dimer conversion. The authors push this equilibrium towards dimeric UNC-104(1-653) motors solely by introducing a point mutation into the coiled-coil domain and ultimately unleash a robust processivity of the UNC-104 dimer. The authors find that the same mechanistic concept does not apply to the KLP-6 kinesin-3 motor, suggesting an alternative activation mechanism of the KLP-6 that remains to be resolved. The present study encourages further dissection of the kinesin-3 motors with the goal of uncovering the main factors needed to overcome the 'self-inflicted' deactivation.

---

## [Author Response]

The following is the authors’ response to the original reviews.

**Public Reviews:**

**Reviewer #1 (Public Review):**
The regulation of motor autoinhibition and activation is essential for efficient intracellular transport. This manuscript used biochemical approaches to explore two members in the kinesin-3 family. They found that releasing UNC-104 autoinhibition triggered its dimerization whereas unlocking KLP-6 autoinhibition is insufficient to activate its processive movement, which suggests that KLP-6 requires additional factors for activation, highlighting the common and diverse mechanisms underlying motor activation. They also identified a coiled-coil domain crucial for the dimerization and processive movement of UNC-104. Overall, these biochemical and single-molecule assays were well performed, and their data support their statements. The manuscript is also clearly written, and these results will be valuable to the field.

Thank you very much!

Ideally, the authors can add some in vivo studies to test the physiological relevance of their in vitro findings, given that the lab is very good at worm genetic manipulations. Otherwise, the authors should speculate the in vivo phenotypes in their Discussion, including E412K mutation in UNC-104, CC2 deletion of UNC-104, D458A in KLP-6.

1. We have shown the phenotypes unc-104(E412K) mutation in *C. elegans* (Niwa et al., Cell Rep, 2016) and described about it in Discussion. The mutant worm showed overactivation of the UNC-104-dependent axonal transport, which is consistent with our biochemical data showing that UNC-104(1-653)(E412K) is prone to form a dimer and more active than wild type.

1. It has been shown that L640F mutation induces a loss of function phenotype in *C. elegans* (Cong et al., 2021). The amount of axonal transport is reduced in unc-104(L640F) mutant worms. L640 is located within the CC2 domain. To show the importance of CC2-dependent dimerization in the axonal transport in vivo, we biochemically investigated the impact of L640F mutation.

By introducing L640F into UNC-104(1-653)(E412K), we performed SEC analysis. The result shows that UNC-104(1-653)(E412K,L640F) failed to form stable dimers despite the release of their autoinhibition (Figure 7—figure supplement 1). This result strongly suggests the importance of the CC2 domain in the axonal transport in vivo. Based on the result, we discussed it in Discussion.

1. Regarding KLP-6(D458A), we need a genetic analysis using genome editing and we would like to reserve it for a future study. We speculate that the D458A mutation could lead to an increase in transport activity in vivo similar to unc-104(E412K). This is because the previous study have shown that wild-type KLP-6 was largely localized in the cell body, while KLP-6(D458A) was enriched at the cell periphery in the N2A cells (Wang et al., 2022). We described it in Discussion.

While beyond the scope of this study, can the author speculate on the candidate for an additional regulator to activate KLP-6 in *C. elegans*?

The heterodimeric mechanoreceptor complex, comprising LOV-1 and PKD-2, stands as potential candidates for regulating KLP-6 dimerization. We speculate the heterodimerization property is suitable for the enhancement of KLP-6 dimerization. On the other hand, it's noteworthy that KLP-6 can undergo activation in Neuro 2a cells upon the release of autoinhibition (Wang et al., 2022). This observation implies the involvement of additional factors which are not present in sf9 cells may be able to induce dimerization. Post-translational modifications would be one of the candidates. We discussed it in Discussion.

The authors discussed the differences between their porcine brain MTs and chlamydonomas axonemes in UNC-104 assays. However, the authors did not really retest UNC-104 on axonemes after more than two decades, thereby not excluding other possibilities.

We thought that comparing different conditions used in different studies is essential for the advancement of the field of molecular motors. Therefore, we newly performed single-molecule assay using Chlamydomonas axonemes and compared the results with brain MTs (Figure 5—figure supplement 2). Just as observed in the study by Tomoshige et al., we were also unable to observe the processive runs of UNC-104(1-653) on Chlamydomonas axonemes (Figure 5—figure supplement 2A). Furthermore, we found that the landing rate of UNC-104(1-653) on Chlamydomonas axonemes was markedly lower in comparison to that on purified porcine microtubules (Figure 5—figure supplement 2B).

**Reviewer #1 (Recommendations For The Authors):**
More discussion as suggested above would improve the manuscript.

We have improved our manuscript as described above.

**Reviewer #2 (Public Review):**
The Kinesin superfamily motors mediate the transport of a wide variety of cargos which are crucial for cells to develop into unique shapes and polarities. Kinesin-3 subfamily motors are among the most conserved and critical classes of kinesin motors which were shown to be self-inhibited in a monomeric state and dimerized to activate motility along microtubules. Recent studies have shown that different members of this family are uniquely activated to undergo a transition from monomers to dimers.Niwa and colleagues study two well-described members of the kinesin-3 superfamily, unc104 and KLP6, to uncover the mechanism of monomer to dimer transition upon activation. Their studies reveal that although both Unc104 and KLP6 are both self-inhibited monomers, their propensities for forming dimers are quite different. The authors relate this difference to a region in the molecules called CC2 which has a higher propensity for forming homodimers. Unc104 readily forms homodimers if its self-inhibited state is disabled while KLP6 does not.The work suggests that although mechanisms for self-inhibited monomeric states are similar, variations in the kinesin-3 dimerization may present a unique form of kinesin-3 motor regulation with implications on the forms of motility functions carried out by these unique kinesin-3 motors.

Thank you very much!

**Reviewer #2 (Recommendations For The Authors):**
The work is interesting but the process of making constructs and following the transition from monomers to dimers seems to be less than logical and haphazard. Recent crystallographic studies for kinesin-3 have shown the fold and interactions for all domains of the motor leading to the self-inhibited state. The mutations described in the manuscript leading to disabling of the monomeric self-inhibited state are referenced but not logically explained in relation to the structures. Many of the deletion constructs could also present other defects that are not presented in the mutations. The above issues prevent wide audience access to understanding the studies carried out by the authors.

We appreciate this comment. We improved it as described bellow.

Suggestions: Authors should present schematic, or structural models for the self-inhibited and dimerized states. The conclusions of the papers should be related to those models. The mutations should be explained with regard to these models and that would allow the readers easier access. Improving access to the readers in and outside the motor field would truly improve the impact of the manuscript on the field.

The structural models illustrating the autoinhibited state have been included, accompanied by an explanation of the correlation between the mutations and these structures in the figure legend. Additionally, schematic models outlining the dimerization process of both UNC-104 and KLP-6 have been provided in Figure 7—figure supplement 2 to enhance reader comprehension of the process.

**Reviewer #3 (Public Review):**
In this work, Kita et al., aim to understand the activation mechanisms of the kinesin-3 motors KLP-6 and UNC-104 from *C. elegans*. As with many other motor proteins involved in intracellular transport processes, KLP-6 and UNC-104 motors suppress their ATPase activities in the absence of cargo molecules. Relieving the autoinhibition is thus a crucial step that initiates the directional transport of intracellular cargo. To investigate the activation mechanisms, the authors make use of mass photometry to determine the oligomeric states of the full-length KLP-6 and the truncated UNC-104(1-653) motors at sub-micromolar concentrations. While full-length KLP-6 remains monomeric, the truncated UNC-104(1-653) displays a sub-population of dimeric motors that is much more pronounced at high concentrations, suggesting a monomer-to-dimer conversion. The authors push this equilibrium towards dimeric UNC-104(1-653) motors solely by introducing a point mutation into the coiled-coil domain and ultimately unleashing a robust processivity of the UNC-104 dimer. The authors find that the same mechanistic concept does not apply to the KLP-6 kinesin-3 motor, suggesting an alternative activation mechanism of the KLP-6 that remains to be resolved. The present study encourages further dissection of the kinesin-3 motors with the goal of uncovering the main factors needed to overcome the 'self-inflicted' deactivation.

Thank you very much!

**Reviewer #3 (Recommendations For The Authors):**
126-128: It is surprising that surface-attachment does not really activate the full-length KLP6 motor (v=48 {plus minus} 42 nm/s). Can the authors provide an example movie of the gliding assay for the FL KLP6 construct? Gliding assays are done by attaching motors via their sfGFP to the surface using anti-GFP antibodies. Did the authors try to attach the full-length KLP-6 motor directly to the surface? If the KLP-6 motor sticks to the surface via its (inhibitory) C-terminus, this attachment would be expected to activate the motor in the gliding assay, ideally approaching the in vivo velocities of the activated motor.

We have included an example kymograph showing the gliding assay of KLP-6FL (Figure 1—figure supplement 1A). When we directly attached KLP-6FL to the surface, the velocity was 0.15 ± 0.02 µm/sec (Figure 1—figure supplement 1B), which is similar to the velocity of KLP-6(1-390). While the velocity observed in the direct-attachment condition is much better than those observed in GFP-mediated condition, the observed velocity remains considerably slower than in vivo velocities. Firstly, we think this is because dimerization of KLP-6 is not induced by the surface attachment. Previous studies have shown that monomeric proteins are generally slower than dimeric proteins in the gliding assay (Tomishige et al., 2002). These are consistent with our observation that KLP-6 remains to be monomeric even when autoinhibition is released. Secondly, in vitro velocity of motors is generally slower than in vivo velocity.

156-157: It seems that the GCN4-mediated dimerization induces aggregation of the KLP6 motor domains as seen in the fractions under the void volume in Figure 3B (not seen with the Sf9 expressed full-length constructs, see Figure 1B). Also, the artificially dimerized motor construct does not fully recapitulate the in vivo velocity of UNC-104. Did the authors analyze the KLP-6(1-390)LZ with mass photometry and is it the only construct that is expressed in *E. coli*?

KLP-6::LZ protein is not aggregating. We have noticed that DNA and RNA from *E. coli* exists in the void fraction and they occasionally trap recombinant kinesin-3 proteins in the void fraction. To effectively remove these nucleic acids from our protein samples, we employed streptomycin sulfate as a purification method (Liang et al., Electrophoresis, 2009). Please see Purification of recombinant proteins in Methods. In the size exclusion chromatography analysis, we observed that KLP-6(1-393)LZ predominantly eluted in the dimer fraction (New Figure 3). Subsequently, we reanalyzed the motor's motility using a total internal reflection fluorescence (TIRF) assay, as shown in the revised Figure 3. Even after these efforts, the velocity was not changed significantly.The velocity of KLP-6LZ is about 0.3 µm/sec while that of cellular KLP-6::GFP is 0.7 µm/sec (Morsci and Barr, 2011). Similar phenomena, "slower velocity in vitro", has been observed in other motor proteins.

169: In Wang et al., (2022) the microtubule-activated ATPase activities of the mutants were measured in vitro as well, with the relative activities of the motor domain and the D458A mutant being very similar. The D458A mutation is introduced into the full-length motor in Wang et al., while in the present work, the mutation is introduced into the truncated KLP-6(1-587) construct. Can the authors explain their reasoning for the latter?

(1) Kinesins are microtubule-stimulated ATPases. i.e. The ATPase activity is induced by the binding with a microtubule.

(2) Previous studies have shown that the one-dimensional movement of the monomeric motor domain of kinesin-3 depends on the ATPase activity even when the movement does not show clear plus-end directionality (Okada et al., Science, 1998).

(3) While KLP-6(1-587) does not bind to microtubules, both KLP-6(1-390) ( = the monomeric motor domain) and KLP-6(1-587)(D458A) similarly bind to microtubules and show one dimensional diffusion on microtubules (Figure 4E and Figure 2—figure supplement 1B).

Therefore, the similar ATPase activities of the motor domain( = KLP-6(1-390)) and KLP-6(D458A) observed by Wang et al. is because both proteins similarly associate with and hydrolyze ATP on microtubules, which is consistent with our observation. On the other hand, because KLP-6(wild type) cannot efficiently bind to microtubules, the ATPase activity is low.

Can the authors compare the gliding velocities of the KLP-6(1-390)LZ vs KLP-6(1-587) vs KLP-6(1-587)(D458A) constructs to make sure that the motors are similarly active?

We conducted a comparative analysis of gliding velocities involving KLP-6(1-390), KLP-6(1-587), and KLP-6(1-587)(D458A) (Figure 1—figure supplement 1C). We used KLP-6(1-390) instead of KLP-6(1-390)LZ, aligning with the protein used by Wang et al.. We demonstrated that both KLP-6(1-587) and KLP-6(1-587) (D458A) exhibited activity levels comparable to that of KLP-6(1-390). The data suggests that the motor of all recombinant proteins are similarly active.

Please note that, unlike full length condition (Figure 1D and Figure 1—figure supplement 1A and S1B), the attachment to the surface using the anti-GFP antibody can activates KLP-6(1-587). The data suggests that, due to the absence of coverage by the MBS and MATH domain (Wang et al., Nat. Commun., 2022), the motor domain of KLP-6(1-587) to some extent permits direct binding to microtubules under gliding assay conditions.

Are the monomeric and dimeric UNC-104(1-653) fractions in Figure 5B in equilibrium? Did the authors do a re-run of the second peak of UNC-104(1-653) (i.e. the monomeric fraction with ~100 kDa) to assess if the monomeric fraction re-equilibrates into a dimer-monomer distribution?

We conducted a re-run of the second peak of UNC-104(1-653) and verified its re-equilibration into a distribution of dimers and monomers after being incubated for 72 hours at 4°C (Figure 5—figure supplement 1).

UNC-104 appears to have another predicted coiled-coiled region around ~800 aa (e.g. by NCoils) that would correspond to the CC3 in the mammalian homolog KIF1A. This raises the question if the elongated UNC-104(1-800) would dimerize more efficiently than UNC-104(1-653) (authors highlight the sub-population of dimerized UNC-104(1-653) at low concentrations in Figure 5C) and if this dimerization alone would suffice to 'match' the UNC-104(1-653)E412K mutant (Figure 5D). Did the authors explore this possibility? This would mean that dimerization does not necessarily require the release of autoinhibition.

We have tried to purify UNC-104(1-800) and full-length UNC-104 using the baculovirus system. However, unfortunately, the expression level of UNC-104(1-800) and full length UNC-104 was too low to perform in vitro assays even though codon optimized vectors were used. Instead, we have analyzed full-length human KIF1A. We found that full-length KIF1A is mostly monomeric, not dimeric (Please look at the Author response image 1). The property is similar to UNC-104(1-653) (Figure 5A-C). Therefore, we think CC3 does not strongly affect dimerization of KIF1A, and probably its ortholog UNC-104. Moreover, a recent study has shown that CC2 domain, but not other CC domains, form a stable dimer in the case of KIF1A (Hummel and Hoogenraad, JCB, 2021). Given the similarity in the sequence of KIF1A and UNC-104, we anticipate that the CC2 domain of UNC-104 significantly contributes to dimerization, potentially more than other CC domains. We explicitly describe it in the Discussion in the revised manuscript.

**Author response image 1. sa4fig1:** Upper left, A representative result of size exclusion chromatography obtained from the analysis of full-length human KIF1A fused with sfGFP. Upper right, A schematic drawing showing the structure of KIF1A fused with sfGFP and a result of SDS-PAGE recovered from SEC analysis. Presumable dimer and monomer peaks are indicated.Lower left, Presumable dimer fractions in SEC were collected and analyzed by mass photometry. The result confirms that the fraction contains considerable amount of dimer KIF1A.Lower right, Presumable monomer fractions were collected and analyzed by mass photometry. The result confirms that the fraction mainly consists of monomer KIF1A.Note that these results obtained from full-length KIF1A protein are similar to those of UNC-104(1-653) protein shown in Figure 5A-C.